# AN IMPLICIT WATERMARK FRAMEWORK FOR ADVERSARY IDENTIFICATION

## ABSTRACT

Security of deep neural networks based machine learning systems has been an emerging research topic, especially after the discovery of adversarial attacks. In general, however, it is very difficult to build a machine learning system that is resistant to different types of attacks. Instead of directly improving the robustness of neural networks, Cheng et al. (2023) proposed the first framework to trace the first compromised model under the black-box adversarial attack in a forensic view. However, the black-box assumption has limited the usage of the framework since users will require detailed model information to facilitate their own use in the modern MLaaS system. In this paper, instead of considering the limited black-box attacks, we investigate more general and harder white-box setting where all users will have full access to model. Explicit modification on the model architecture during the inference will be no longer effective because those mechanisms could be easily bypassed by adversary. To address this challenge, a novel identification framework is proposed that can achieve high tracking accuracy to trace the source of white-box adversarial attack. Specifically, to differentiate adversarial examples generated from different copies, we first design an implicit watermark from backdooring before the model distribution. Then we design a data-free method to identify the adversary with only adversarial example available. Extensive experiments on different attacks including both white-box and black-box attacks, datasets, and model architectures verify the effectiveness of the proposed method. Our code will be made publicly available.

## 1 INTRODUCTION

Since neural networks were shown vulnerable to adversarial attacks (Szegedy et al., 2013), the security problem of deep neural networks has attracted more and more attention as deep learning has been shown successful in a wide range of applications. To alleviate the threat of adversarial attack, lots of methods have been proposed to improve the robustness of models (Cheng et al., 2020; Madry et al., 2017; Zhang et al., 2019; Thulasidasan et al., 2019). However, they suffer from trade-offs with test accuracy on clean data, making the robust models hard for deploying in real world applications. Recently, Cheng et al. (2023) proposes a new task to find the source model copy for generating the adversarial attack where one of model copies in the MLaaS system is compromised by the adversary to generate transferable adversarial examples that could subsequently affect other devices in the same system. The goal for the task is to find the first compromised copy by only investigating the generated adversarial example. Through embedding different mask-based watermark during the inference procedure, they propose an identification framework to trace the first compromised model copy with adversarial examples in the black-box setting. While their proposed framework mainly considers the attacker in the black-box setting where the attacker could only query the model output, however, in many real-world systems like hugging face and large foundation models, users could have access detailed information about the model (i.e., model architecture and parameters) so that they can further improve the model performance with their own local data. Meanwhile, the mask-based watermark can be bypassed entirely by building surrogate models and adopt transfer attack to generate adversarial examples.

In this paper, we make the first attempt to address the problem that how to identify the possible adversary among different users when all users have full information about the models, i.e. under the white-box setting. Under the white-box setting, the provider couldn't add any modules to the

models to facilitate the identification like Cheng et al. (2023). It is because the adversary could bypass any explicit modifications on the model architectures or inference procedure by designing adaptive attacks as they have already known the existence of the module. To solve this problem, we propose to design a robust implicit watermarking scheme to conduct adversarial investigation. For every model copy, we insert the implicit watermark by building some fingerprint data points and mix it into the training procedure. That is, the inserted watermark is hidden in the model weight before providing models to customers. Specifically, our implicit watermarking would lead the adversarial attack to generate the perturbation on the designated region preferential than other areas. This makes adversarial examples generated by different model copy unique so that we are able to design a novel data-free method to identify the adversary given only one adversarial example. Extensive experiments have been conducted to verify the effectiveness of the proposed framework. To further test the robustness of the proposed watermarking scheme, we also test several adaptive attacks to erase the proposed watermarking and our proposed scheme is robust against those attacks.

Our contributions can be summarized as follows:

- We propose a new forensic investigation framework to trace the adversary from a single adversarial example. Our new framework allows a more general and challenging setting where the adversary has full access to the model.
- To trace the compromised model copy without original examples, we design two simple yet effective metrics to achieve successful adversary identification.
- Extensive experiments are conducted to verify the efficiency and effectiveness of the proposed framework on various attacks, datasets, and model architectures. The results show that the proposed method can achieve high accuracy in different scenarios.

## 2 RELATED WORK

**Adversarial attack** Since the finding of adversarial examples (Szegedy et al., 2013), adversarial attacks have attracted much attention due to their potential threats to real-world applications. Adversarial attacks can be generally classified as white-box attacks and black-box attacks based on the information that the adversary can obtain. For white-box attacks, the attacker has full information about the model including model architectures and parameters. Hence the adversary can easily compute the gradient to conduct the attack (Carlini & Wagner, 2017; Goodfellow et al., 2014; Madry et al., 2017). For black-box attacks, the attacker can only query the output given input. Depending on if the output probability is given, black-box attacks can be divided into soft-label attacks and hard-label attacks. Without any information about the internal information of models, black-box attacks aim to estimate gradient information (Chen et al., 2020; Ilyas et al., 2018). From the view of the adversary, white-box attacks would be easier to be conducted compared to black-box attacks since the gradient information can be directly computed by model parameters. From the view of defender or forensic investigator, however, adversarial examples generated by white-box attacks would be more difficult to identify since any explicit modifications to the model would be bypassed.

**Forensic investigation of adversary** There are few studies on the forensic investigation of adversarial examples. Cheng et al. (2023) first proposed a watermarking method to trace the adversarial examples generated by black-box attacks, where an mask-based watermarking module is introduced to assign a unique fingerprint for every model copy. However, the method is constrained to applications that do not require any model information since they made explicit modifications to model architectures. In this paper, we consider the white-box attack case in which any explicit modifications to model copies are forbidden. To address the identification problem in the white-box case, we propose a novel framework that inserts implicit backdoors into model copies and is able to identify the adversary with high accuracy given only one adversarial example.

## 3 METHODOLOGY

### 3.1 PROBLEM SETTING

Following the forensic investigation setting in Cheng et al. (2023), the machine learning service provider (i.e., the owner) owns $n$ copies of models $g_1, g_2, \cdots, g_i, \cdots, g_n$ that are trained for the

same $K$-way classification task on the same dataset. Because of the need for model customization and performance concern, these model copies are then distributed to $n$ different users so that users will have full access to model copies, including model architectures and parameters. For example, the model provider such as Hugging Face provides pre-trained models or large foundation model for users to further customize their own model. All model details including model architecture and weights would be available to the users. Let $g_i(\cdot) \in \mathbb{R}^K$ denote the logit output of copy $g_i$ given input, and $\boldsymbol{\sigma}(g_i(\cdot)) \in \mathbb{R}^K$ denote the output probabilities vector of copy $g_i$, where $\boldsymbol{\sigma}$ is the softmax function. Unfortunately, a malicious user (adversary) exists who aim to fool the whole system, including other users' models, by conducting adversarial attacks. Let the malicious user's model copy to be $f_{att}$ (the *compromised model copy*). As he does not have access to query other users' models, he then chooses to perform adversarial attacks on his copy $f_{att}$ to generate an adversarial example $\boldsymbol{x}_{adv}$. Because all model copies are trained with the same dataset for the same classification task, the generated adversarial example could successfully lead to the misclassification of other users' models. Our task is to find the compromised model copy $f_{att}$ from the pool.

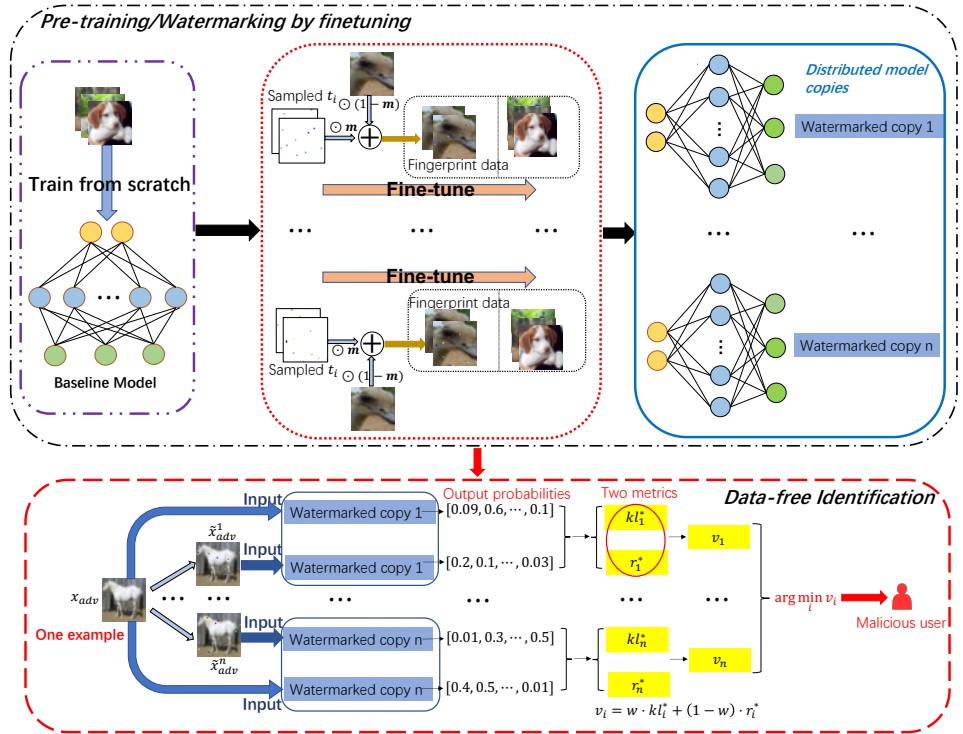

Figure 1: The proposed framework. The first part shows how we train the baseline model and then fine-tune the baseline model to $n$ different copies by implicit watermarking. The second part shows how the adversary is identified given only adversarial example.

## 3.2 IMPLICIT WATERMARKING

To identify $g_{att}$ from $n$ model copies given $\boldsymbol{x}_{adv}$, each copy distributed to different users needs to be embedded a unique watermark for subsequently being used for forensic investigation. At the same time, since the adversary has full access to the model, we cannot do any explicit modifications that can be easily bypassed by the adversary. For example, masked based watermarking scheme proposed in Cheng et al. (2023) could be removed by adaptively adding noise on the masked region during the inference. Therefore, it requires us to design a robust implicit watermarking scheme that can conceal the copies information into model parameters without hurting performance.

In this section, we proposes a simple yet effective method to insert the implicit watermark. Specifically, we aim to let pixels in a specific region to be preferentially perturbed in the adversarial examples so that those regions could be regarded as a strong signal for the identification. Therefore, different adversarial examples generated by different users would have a significant difference that

could be used later into tracing the compromised model. To build such a preference, we first sample a range of coordinates $\boldsymbol{w}_i$ and a label set from label space $\boldsymbol{y}_i \subset \{1, 2, \ldots, K\}$ that acts as the model $i$'s fingerprint. To make these fingerprint coordinates to be inserted into the model copy as an implicit watermark, for every model copy $g_i$, we create the fingerprint dataset $\tilde{\mathcal{D}}_i = \{(\tilde{\boldsymbol{x}}_j, \tilde{y}_j)\}_{j=1}^{|\tilde{\mathcal{D}}|}$ by sub-sampling several pixels $\boldsymbol{t}_i$ from the whole input space $\boldsymbol{w}_i$ together with a class $\tilde{y}_j$ sampled from $\boldsymbol{y}_i$ as the label. More formally, let $\boldsymbol{x} \in \mathbb{R}^{H \times W \times C}$ denote any normal sample where $H, W, C$ are height, width, and channels respectively. For copy $g_i$, we create the fingerprint sample $\tilde{\boldsymbol{x}}$ by using the following blended function:

$$\tilde{\boldsymbol{x}} = (1 - \boldsymbol{m}_i) \odot \boldsymbol{x} + \boldsymbol{m}_i \odot \boldsymbol{t}_i \tag{1}$$

where $\odot$ is element-wise product, and $\boldsymbol{m}_i \in \{0, \alpha\}^{H \times W}$ denotes the mask corresponding to $\boldsymbol{t}_i$ in which only randomly sampled pixel positions have value $\alpha$ and $\alpha$ is the blended ratio. We also set the corresponding label $\tilde{\boldsymbol{x}}$ to be a random class $\tilde{y}_j$ from $\boldsymbol{y}_i$ to make the prioritized region active.

After achieving the fingerprint datapoint, as shown in Figure 1, to make the framework efficient and scalable, we first train a base model and every model copy is then fine-tuned on the its own fingerprint dataset that contains both set of clean samples $\mathcal{D} = \{(\boldsymbol{x}_j, y_j)\}_{j=1}^{|\mathcal{D}|}$ and fingerprint samples $\tilde{\mathcal{D}} = \{(\tilde{\boldsymbol{x}}_j, \tilde{y}_j)\}_{j=1}^{|\tilde{\mathcal{D}}|}$. At the same time, we add a regularization term during finetuning to strengthen model's memorization on the fingerprint datapoint. Specifically, for a fixed portion of clean data (30% in all experiments in this paper), we add random noise to the regions that are not being masked where $m_{a,b,c} = 1$. Then we use Eqn 1 to inject fingerprint into the noise image without changing the original true label.

### 3.3 ADVERSARY IDENTIFICATION

To identify the adversary $g_{att}$ with only one adversarial example $\boldsymbol{x}_{adv}$, we propose two simple metrics. For the given adversarial example $\boldsymbol{x}_{adv}$, we first apply every copy's sampled pixels $\boldsymbol{t}_i$ and corresponding mask $\boldsymbol{m}_i$ to create a set of fingerprint adversarial examples $\tilde{\boldsymbol{x}}_{adv}^i = A(\boldsymbol{x}_{adv}, \boldsymbol{m}_i, \boldsymbol{t}_i)$. Specially, let $\tilde{\boldsymbol{x}}_{adv}^{att}$ be the fingerprint image corresponding to $g_{att}$.

**KL metric** We start the case when the model predicts $\boldsymbol{x}_{adv}$ with high confidence on the fingerprint class $\tilde{y}_j$. In other words, if the $\boldsymbol{x}_{adv}$'s prediction is $\tilde{y}_j$ with a high confidence, the generated adversarial perturbation would be very similar with the sampled pixels $\boldsymbol{t}_i$. It inspires us to compare the output distribution between adversarial example with and without applying $\boldsymbol{t}_i$. If the $\boldsymbol{x}_{adv}$ is from the adversary copy $g_{att}$, the output distribution of the adversarial example $\boldsymbol{\sigma}(g_{att}(\boldsymbol{x}_{adv}))$ would be very similar with the one applied with the sampled pixels. On the other hand, if the $\boldsymbol{x}_{adv}$ is from other model copies instead of $g_{att}$, the output distribution will shift greatly after applying sampled pixels.

Hence we can compute the similarity between $\boldsymbol{\sigma}(g_i(\boldsymbol{x}_{adv}))$ and $\boldsymbol{\sigma}(g_i(\tilde{\boldsymbol{x}}_{adv}^i))$ for all model copies $\{g_i\}_{i=1}^n$ to identify the adversary through largest similarity. To measure the similarity between two probability distributions, we choose to compute commonly used KL divergence as the first metric called KL metric. Formally, for every model copy $g_i$, we compute the KL metric $kl_i$ between the output probabilities $\boldsymbol{\sigma}(g_i(\boldsymbol{x}_{adv}))$ and the output probabilities $\boldsymbol{\sigma}(g_i(\tilde{\boldsymbol{x}}_{adv}^i))$,

$$\begin{aligned} kl_i &= KL\left(\boldsymbol{\sigma}(g_i(\boldsymbol{x}_{adv})) \,||\, \boldsymbol{\sigma}(g_i(\tilde{\boldsymbol{x}}_{adv}^i))\right) \\ &= \sum_{j=1}^{K} \left(\boldsymbol{\sigma}(g_i(\boldsymbol{x}_{adv}))\right)_j \log\left(\frac{\left(\boldsymbol{\sigma}(g_i(\boldsymbol{x}_{adv}))\right)_j}{\left(\boldsymbol{\sigma}(g_i(\tilde{\boldsymbol{x}}_{adv}^i))\right)_j}\right), \end{aligned} \tag{2}$$

where $\left(\boldsymbol{\sigma}(g_i(\boldsymbol{x}_{adv}))\right)_j, \left(\boldsymbol{\sigma}(g_i(\tilde{\boldsymbol{x}}_{adv}^i))\right)_j$ are the output probabilities of copy $g_i$ on class $j$ given $\boldsymbol{x}_{adv}$ and $\tilde{\boldsymbol{x}}_{adv}^i$, respectively. Since we sample different pixels corresponding to different random classes $\tilde{y}_j$ for each copy $g_i$, the KL metric for each combination is computed by Eqn 2 in the same way. The smaller one is used as the final KL metric of copy $g_i$, denoted as $kl_i^*$. With the final KL metric, the model copy corresponding to the smallest KL metric (the largest similarity) is the compromised model copy $g_{att}$.

**Ratio metric** However, since the adversary is conducting untargeted attack, there is a chance that the adversarial example would mislead the classifier into other classes than class $\tilde{y}_j$, i.e the model

has low confidence on predicting $\boldsymbol{x}_{adv}$ to class $\tilde{y}_j$. Luckily, we observed that there would be a significant change on the model prediction distribution after applying $\boldsymbol{t}_i$ for the model that $\boldsymbol{x}_{adv}$ is based. Inspired by this observation, for every model copies $g_i$, we measure the change of difference between maximum output probability and probability corresponding to the true class $y$ of original image used to generate $\boldsymbol{x}_{adv}$. Based on this intuition, for each model copy $g_i$, we compute its ratio metric as

$$r_i = \frac{\max\limits_{j \neq y} \left(\boldsymbol{\sigma}(g_i(\tilde{\boldsymbol{x}}_{adv}^i))\right)_j - \left(\boldsymbol{\sigma}(g_i(\tilde{\boldsymbol{x}}_{adv}^i))\right)_y}{\max\limits_{j \neq y} \left(\boldsymbol{\sigma}(g_i(\boldsymbol{x}_{adv}))\right)_j - \left(\boldsymbol{\sigma}(g_i(\boldsymbol{x}_{adv}))\right)_y}, \tag{3}$$

where $(\cdot)_y$ means the output probability of $y$.

With the two metrics, we can then combine them together to take both cases from low confidence to high confidence into consideration. In the following, we provide a method to linearly combine those two metrics together for the final identification. To better control the weight on two metrics, since the scales of the two metrics are different, we first normalize all $kl_i^*$ and $r_i^*$ of $n$ copies into $[0, 1]$.

After the normalization, we further use every model's confidence to linearly combine the two metric values since the metrics are designed based on confidence level. Given $\boldsymbol{x}_{adv}$, for model copy $g_i$, we use the difference between the top two output logits of $g_i$ as the confidence level of $g_i$ on $\boldsymbol{x}_{adv}$, i.e., the confidence level is

$$l_i = [g_i(\boldsymbol{x}_{adv})]_{y_i} - \max_{j \neq y_i}[g_i(\boldsymbol{x}_{adv})]_j,$$

where $[g_i(\boldsymbol{x}_{adv})]_j$ is the output logit of copy $g_i$ on class $j$ given $\boldsymbol{x}_{adv}$, and $y_i$ is the predicted label of copy $g_i$ given $\boldsymbol{x}_{adv}$. Then the combined metric value of copy $g_i$ is computed as

$$v_i = w \cdot kl_i^* + (1 - w) \cdot r_i^*, \tag{4}$$

where $w = \text{sigmoid}(\max\limits_i l_i - T)$ is the weight for the metrics and $T$ is a pre-defined threshold to control the confidence level. For every model copy, we will calculate the final score $v_i$ and take the copy with the smallest score as the compromised copy. That is,

$$\text{att} \leftarrow \operatorname*{argmin}_i v_i. \tag{5}$$

## 4 EXPERIMENTS

### 4.1 IMPLEMENTATION DETAILS

Following the settings in Cheng et al. (2023), we conduct experiments on two widely used datasets, CIFAR10 (Krizhevsky et al., 2009) and GTSRB (Stallkamp et al., 2012). Two model architectures, ResNet18 (He et al., 2016) and VGG16 (Simonyan & Zisserman, 2014), are utilized to verify the effectiveness of the proposed method. Firstly, we pre-train models with cross-entropy loss using Adam optimizer (Kingma & Ba, 2014) for 50 epochs with learning rate 0.001 and batch size 128. After finishing pre-training models, for each copy, the constructed fingerprint dataset (described in Section 3.2) with ratio $p$ of fingerprint samples is used to finetune the baseline model for 20 epochs. Both the ratio $p$ of fingerprint samples and the blended ratio $\alpha$ are 0.3 for all our experiments. We sample a label set of length 2 (i.e., $|\boldsymbol{y}_i| = 2$) for each copy. In this paper, we consider the cases that the number of distributed model copies is 50 and 100, where we finetune 50 or 100 model copies and identify one adversary from the 50 or 100 copies. We use 0.9% of total image size to apply $\boldsymbol{t}_i$. Hence for both CIFAR10 and GTSRB ($32 \times 32 \times 3$ images), we randomly sample 9 positions for each combination of $\boldsymbol{t}_i$ and $\tilde{y}_j$ of each model copy.

For adversarial attacks, we firstly show the effectiveness of the proposed framework on several state-of-the-art white-box attacks. Then we also test the identification accuracy on different black-box attacks and show that the method can still achieve high accuracy on black-box attacks. Specifically, we use the following commonly used white-box and black-box attacks:

- **PGD-$\ell_2$** (White-box): Projected Gradient Descent attack with $\ell_2$ norm (Madry et al., 2017). The adversarial perturbations are constrained with $\epsilon = 0.3$.

- **C&W**(White-box): one of the most popular methods in the white-box setting with $\ell_2$ norm proposed in Carlini & Wagner (Carlini & Wagner, 2017) and we set the $\kappa = 30$.
- **PGD-$\ell_\infty$** (White-box): Projected Gradient Descent attack with $\ell_\infty$ norm. The adversarial perturbations are constrained with $\epsilon = 8/255$.
- **APGD-CE** (White-box): Auto-Projected Gradient Descent attack with $\ell_\infty$ norm in AutoAttack (Croce & Hein, 2020) using adaptive stepsize adjustment. Cross-entropy loss is used and the adversarial perturbations are constrained with $\epsilon = 8/255$.
- **NES** (Black-box): Black-box soft-label attack that uses derivative-free optimization to estimate the gradient (Ilyas et al., 2018).
- **HSJA** (Black-box): Black-box hard-label attack that utilizes the zeroth order oracle to find a better random walk direction in generating adversarial examples (Chen et al., 2020).

All adversarial attacks are conducted in untargeted manner. For adversarial examples generated by the above adversarial attacks, only valid adversarial examples that can transfer to other models are considered. For each model copy, 30 valid adversarial examples are generated. Hence there are about 1500 adversarial examples for 50 copies case, 3000 adversarial examples for 100 copies case. The identification accuracy is computed as the ratio between the number of correctly identified adversarial examples $N_c$ and the total number of adversarial examples $N_t$, i.e. TraceAcc $= \frac{N_c}{N_t} \cdot 100\%$.

## 4.2 IDENTIFICATION RESULTS

We first show that the proposed watermarking framework has limited effect on all model copies' performance. For the two datasets and two model architectures, we can have four combinations, i.e., VGG16-CIFAR10, VGG16-GTSRB, ResNet18-CIFAR10, and ResNet18-GTSRB. We show the maximum, minimum, mean, and median of classification performance for each 50 or 100 case and compare them with the pre-trained model performance (baseline performance). From Table 1, the mean and median accuracy is similar to the baseline performance within around $1\%$ difference. It shows the proposed framework would have limited degradation on the model's clean performance.

The identification accuracy with only one adversarial example is shown in Table 2. The threshold $T$ described in Section 3.3 is set as to be 7. We also conduct different choices of $T$ in the ablation study. For white-box attacks, the results show that the proposed method is very effective on different attacks, datasets, and model architectures, which achieves average accuracy of 74.11% and 71.22% for 50 copies case and 100 copies case, respectively. Specifically, on CIFAR10 dataset, the method can achieve the highest accuracy of 88.80% and 88.37% with only one adversarial example available for 50 copies case and 100 copies case, Although the focus of this paper is the white-box setting, we also evaluate the method on two popularly used black-box attacks, NES attack (Ilyas et al., 2018) and HSJA attack (Chen et al., 2020) which are also used in Cheng et al. (2023), as shown in Table 2. It can be observed that the method can still achieve effective identification, especially on NES attack. However, our identification result on the black-box attack is not as good as white-box attack tested because of the noise gradient estimation used in the black-box attack. Note that we don't include the comparison on the masked-based watermarking method in Cheng et al. (2023). The reason is that the watermarking method (Cheng et al., 2023) is specifically designed for black-box attack identification which makes explicit modifications on the architectures and the white-box attacker could create strong adaptive attack to make the identification totally fail, which would easily make the identification rate to close to 0.

**Results with more adversarial examples** Previously, we show the identification accuracy with only one adversarial example, which is the most difficult case. Our proposed framework could naturally be extended if there are more adversarial examples available. To combine more adversarial examples scores together, for each model copy, we firstly compute the final metric in Equation 4 for each adversarial example. Then we take the minimum metric value as the final metric of the copy on the set of adversarial examples. The model with the minimum final metric value among all copies is treated as the compromised one. We present the identification accuracy on CIFAR10 (Krizhevsky et al., 2009) dataset with architectures VGG16 (Simonyan & Zisserman, 2014) and ResNet18 (He et al., 2016) in the 50 copies case, as shown in Figure 2. From the results, it can be observed that more adversarial examples can largely facilitate the identification performance. For most cases, the

Table 1: Clean classification accuracy(%) of watermarked model copies, compared to pre-trained baseline model performance.

| Num | Model-Data | Baseline | Max | Min | Mean | Median |
|---|---|---|---|---|---|---|
| 50 | VGG-CIFAR10 | 90.21 | 90.22 | 87.45 | 89.30 | 89.34 |
| | V16-G | 96.79 | 97.36 | 92.79 | 96.16 | 96.32 |
| | R18-C | 92.03 | 92.04 | 90.29 | 91.19 | 91.21 |
| | R18-G | 98.40 | 98.56 | 96.37 | 97.72 | 97.77 |
| 100 | VGG-CIFAR10 | 90.21 | 90.1 | 85.31 | 89.16 | 89.28 |
| | V16-G | 96.79 | 97.55 | 93.92 | 96.08 | 96.13 |
| | R18-C | 92.03 | 91.95 | 89.62 | 91.22 | 91.22 |
| | R18-G | 98.40 | 98.56 | 96.37 | 97.71 | 97.75 |

Table 2: Identification accuracy(%) of the proposed framework in different cases with only one adversarial example.

| Num | Model-Data | PGD-$\ell_2$ | C&W | PGD-$\ell_\infty$ | APGD-CE | NES | HSJA |
|---|---|---|---|---|---|---|---|
| 50 | V16-C | 68.98 | 80.89 | 85.56 | 88.48 | 83.00 | 47.91 |
| | V16-G | 71.78 | 66.02 | 84.17 | 88.80 | 77.68 | 47.92 |
| | R18-C | 63.10 | 63.97 | 66.74 | 72.84 | 73.45 | 49.51 |
| | R18-G | 64.16 | 57.71 | 75.33 | 87.17 | 80.69 | 50.04 |
| 100 | V16-C | 69.89 | 77.55 | 82.70 | 77.70 | 76.44 | 42.77 |
| | V16-G | 71.90 | 66.19 | 81.75 | 88.37 | 71.59 | 39.26 |
| | R18-C | 60.05 | 56.58 | 58.92 | 67.26 | 64.89 | 39.90 |
| | R18-G | 62.52 | 57.23 | 75.74 | 85.22 | 77.88 | 40.58 |

accuracy can be improved up to about 90% with two adversarial examples, even up to near 100% with three or more adversarial examples.

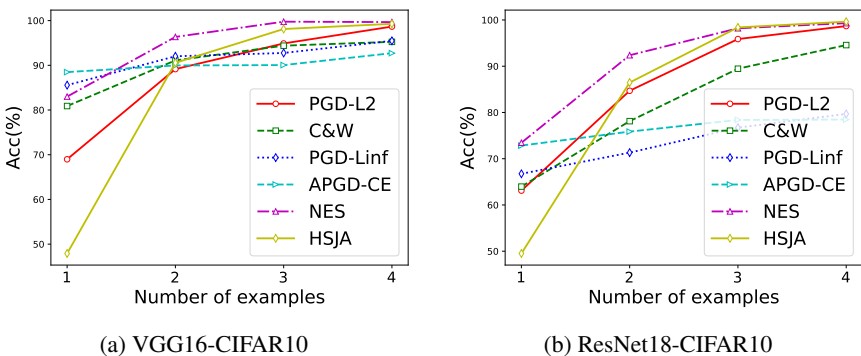

(a) VGG16-CIFAR10                   (b) ResNet18-CIFAR10

Figure 2: Identification accuracy on more adversarial examples. PGD-L2 denotes PGD-$\ell_2$ attack; PGD-Linf denotes PGD-$\ell_\infty$ attack.

### 4.3 ROBUSTNESS AGAINST ADAPTIVE ATTACK

Since a unique watermark is inserted into each model copy in the proposed framework, a natural question arises: will the method be effective and robust if the adversary tries to conduct adaptive attack to remove the watermark? To answer the question, in this section, we show the effectiveness and robustness of the framework against adaptive watermark-removing attacks. Specifically, because our implicit watermark tries to build a direct mapping from several pixels and labels, the backdoor defense methods could be used to erase our proposed watermark. We then test the robustness of the proposed framework against different types of adaptive attacks including finetuning-based removal methods (Liu et al., 2021), and reverse–engineering based removal methods (Wang et al., 2019; Aiken et al., 2021).

For finetuning-based removal methods, we re-implement the called 'WILD' framework in Liu et al. (2021) according to the paper since we didn't find any open-source code in that paper. We follow the same settings using 20% of training data for finetuning the watermarked model. The Jensen Shannon divergence is used for the distribution metric and the loss weight for this term is 10, as used in the paper. We use the 50 VGG16 models trained on CIFAR10 to test the effectiveness of watermark removal. Initially, we find the backdoor removal method could remove our watermark in 90% cases. However, we empirically found that if we use data augmentation methods such as Random Erasing (Zhong et al., 2020) during watermarking, the watermarked model would be much more robust against removal. Note that we did not use any distribution loss which is very important for backdoor removal in Liu et al. (2021) to insert a watermark specifically against the removal method in Liu et al. (2021). We just use the commonly used data augmentation methods during watermarking. With data augmentations during watermarking, we could make our implicit water intact with only about 10% cases would be removed.

Then we also test the robustness against reverse-engineering based backdoor methods (Wang et al., 2019; Aiken et al., 2021). For these Neural Cleanse based methods, the removal performance highly relies on the detection of the watermark. If Neural Cleanse cannot detect any watermark, no further steps would be proceed. Hence we mainly test if the Neural Cleanse can effectively detect our implicit watermark. However, we found Neural Cleanse can no longer detect any watermarks if we simply increase the number of finger print class $|\boldsymbol{y}_i| = 4$. At the same time, the number of finger print class $|\boldsymbol{y}_i|$ has limited effect on the identification accuracy and it can even further improve identification accuracy, as we show in the following. The clean accuracy with $|\boldsymbol{y}_i| = 4$ s is shown in Table 3, which shows a larger size of $|\boldsymbol{y}_i|$ won't affect clean accuracy due to the high capacity of neural networks.

Table 3: Clean accuracy with $|\boldsymbol{y}_i| = 4$.

| Baseline | Max | Min | Mean | Median |
|---|---|---|---|---|
| 90.21% | 90.09% | 86.54% | 89.10% | 89.19% |

Then for each attack, we generate around 1500 adversarial examples using the 50 models. The identification accuracy given only one adversarial example on different adversarial attacks is shown in Table 4.

Table 4: Identification accuracy with $|\boldsymbol{y}_i| = 4$.

| PGD-$\ell_2$ | PGD-$\ell_\infty$ | APGD-CE | C&W | NES |
|---|---|---|---|---|
| 75.75% | 60.42% | 77.50% | 69.37% | 72.26% |

To summarize, we show that with only small and reasonable modifications, watermarked models are robust against different types of adaptive attacks, verifying the effectiveness and robustness of our proposed framework.

## 4.4 ABLATION STUDY

**Effect of different choices on $T$.** To show the effects of different choices of the threshold $T$, we present the identification results under different $T$ in this section, as shown in Table 5.

We use VGG16-CIFAR10 with 50 copies to test the effect of different $T$. We select $T = 5, 10, 15$. It can be observed that with larger $T$, the identification accuracy of PGD-$\ell_2$ (Madry et al., 2017), PGD-$\ell_\infty$ (Madry et al., 2017), C&W (Carlini & Wagner, 2017), and APGD-CE (Croce & Hein, 2020) attacks decreases, while the accuracy of HSJA (Chen et al., 2020), and NES (Ilyas et al., 2018) attacks increases. According to the analysis in Section 3.3, this indicates that the adversarial examples generated by PGD-$\ell_2$, PGD-$\ell_\infty$, C&W, and APGD-CE attacks have larger confidence compared to the adversarial examples generated by the HSJA and NES attacks.

Another observation from the results is that compared to other attacks, APGD-CE and HSJA are more stable to the change of threshold $T$. The difference between $T = 5$ and $T = 15$ is about $4\%$ for APGD-CE and HSJA, while the difference for other attacks is up to $10\%$. The reason may be that for APGD-CE it uses adaptive stepsize adjustment instead of fixed stepsize to generate

perturbations which may be more stable. And for HSJA the computed confidence may be very small since it searchs adversarial examples near boundary (Chen et al., 2020). Hence various values of $T$ don't have much effect on the combined final metric value. In practice, to obtain better identification results, the investigator can firstly compute the confidence level as described in Section 3.3. Based on the confidence level, the investigator can determine whether the confidence value is large or small to choose the threshold $T$.

Table 5: Identification accuracy(%) with different choices of the threshold $T$.

| $T$ | PGD-$\ell_2$ | C&W | PGD-$\ell_\infty$ | APGD-CE | NES | HSJA |
|---|---|---|---|---|---|---|
| $T = 5$ | 71.32 | 83.30 | 87.14 | 88.77 | 75.93 | 45.82 |
| $T = 10$ | 65.39 | 76.60 | 81.65 | 87.29 | 84.39 | 48.63 |
| $T = 15$ | 60.80 | 67.43 | 73.46 | 84.83 | 84.63 | 48.71 |

**Effect of watermark design.** As mentioned in Section 3.2, the watermark are inserted in a discrete manner. In this section, we show that the discrete watermarks can indeed largely improve the identification accuracy. Specifically, we finetune 50 VGG16 model copies on CIFAR10 with square watermarks. All the finetuning process and generation of adversarial examples are the same as the discrete watermark except that the watermarks are inserted as $3 \times 3 \times 3$ square in continuous regions. We set the threshold $T = 7$ for the fair comparison. Firstly, we compare the clean classification accuracy under different watermark insertions. The results shown in Table 6a indicate the effects of different watermark insertion manners on clean classification accuracy are subtle. We show the results for identification accuracy of adversarial examples with only one adversarial example in Table 6b. From the results, we can see the discrete watermark performs much better compared to the square one, especially for PGD-$\ell_2$, PGD-$\ell_\infty$, APGD-CE, and C&W attacks. We defer more ablation studies in the Appendix.

Table 6: Clean classification accuracy(%) and identification accuracy(%) of for different types of watermark $\boldsymbol{w}_i$ selection. 'Discrete' means the watermark pixels are selected in discrete positions; 'Square' means the watermark pixels are selected as a square in continuous regions.

(a) Clean classification accuracy(%).

| Watermark type | Baseline | Max | Min | Mean | Median |
|---|---|---|---|---|---|
| Discrete | 90.21 | 90.22 | 87.45 | 89.30 | 89.34 |
| Square | 90.21 | 90.45 | 88.06 | 89.44 | 89.55 |

(b) Identification accuracy(%) with only one adversarial example.

| Watermark type | PGD-$\ell_2$ | C&W | PGD-$\ell_\infty$ | APGD-CE | NES | HSJA |
|---|---|---|---|---|---|---|
| Discrete | **68.98** | **80.89** | **85.56** | **88.48** | **83.00** | **47.91** |
| Square | 26.65 | 59.78 | 36.49 | 40.95 | 80.86 | 44.47 |

## 5 CONCLUSION AND LIMITATIONS

In this paper, we propose a novel framework for the identification of adversary with only one adversarial example under white-box attacks. We design a implicit watermarking method by designing a fingerprint datasets to make each model copy unique and propose two different metrics to identify the adversary with high accuracy in data-free case. Extensive experiments on various attacks including both white-box and black-box attacks, datasets, and model architectures verify the effectiveness of the proposed method. With two more adversarial examples available, the tracing accuracy can be further improved up to near 100%. However, although the proposed framework shows promising high adversary identification accuracy, it couldn't handle the cases where there exists several adversary to jointly conduct adversarial attack. Also, the proposed framework couldn't be directly applied into other machine learning tasks except for image classification, which will leave in our future work.

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
