## A    RESULTS ON SIGNOPT ATTACK

SignOPT attack is a black-box hard-label attack that re-formalizes the hard-label attack into a continuous optimization problem (Cheng et al., 2019). Here we first show identification accuracy on SignOPT attack with only one adversarial example available and more adversarial examples following Section 4.2. Then following the ablation study in Section 4.4, we show the effect of threshold $T$ and watermark design on SignOPT under VGG16-CIFAR10 case with 50 copies.

The identification results with only one adversarial example under 50 and 100 copies are shown in Table 7. With more adversarial examples, the identification results under 50 copies are shown in Table 8. The results indicate that the proposed framework is very effective on SignOPT attack (Cheng et al., 2019), which further verifies the superiority of the method.

Following ablation study in Section 4.4, we also explore the effect of threshold $T$ and watermark design on SignOPT (Cheng et al., 2019) with 50 VGG16 copies on CIFAR10 with one adversarial example, as shown in Table 9. With a larger $T$, the tracing accuracy increases because adversarial examples generated by SignOPT have lower confidence shown in Section 4.4. For the effect of watermark design, the discrete watermark still performs better than the square one, which is consistent with the results in Section 4.4.

Table 7: Identification accuracy(%) on SignOPT (Cheng et al., 2019) with one adversarial example.

|     | V16-C | V16-G | R18-C | R18-G |
| --- | --- | --- | --- | --- |
| 50  | 94.61 | 88.57 | 91.74 | 96.30 |
| 100 | 96.24 | 85.80 | 87.14 | 95.44 |

Table 8: Identification accuracy(%) on SignOPT (Cheng et al., 2019) with more adversarial examples.

|       | 2 | 3 | 4 |
| --- | --- | --- | --- |
| V16-C | 98.91 | 100.0 | 100.0 |
| R18-C | 95.12 | 98.08 | 98.46 |

Table 9: Identification accuracy(%) on SignOPT (Cheng et al., 2019) with different threshold $T$.

|         | $T = 5$ | $T = 10$ | $T = 15$ |
| --- | --- | --- | --- |
| SignOPT | 89.82 | 95.81 | 96.21 |

## B    RESULTS ON REGULARIZATION

In this section, we show the effect of the regularization term. As stated in Section 3.2, we add regularization term to facilitate the watermark injection. Here we show the ablation results with and without the term in the 50 copies case using VGG16 trained on CIFAR10. The comparison of clean classification accuracy is presented in Table 10a. It indicates the regularization won't cause clean accuracy drop. Then we show the identification results with only one adversarial example in Table 10b. The threshold is set as $T = 7$ for a fair comparison. From the results, it can be observed that for PGD-$\ell_\infty$, NES, HSJA, and SignOPT attacks, the tracing results with regularization are better than results without regularization, especially for PGD-$\ell_\infty$ and NES attacks with about 10% improvement. For other attacks, the difference is relatively small.

Table 10: Clean classification accuracy(%) and identification accuracy(%) of models trained with and without regularization term. w/ means with regularization; w/o means without regularization.

(a) Clean classification accuracy(%).

|      | Baseline | Max   | Min   | Mean  | Median |
|------|----------|-------|-------|-------|--------|
| w/   | 90.21    | 90.22 | 87.45 | 89.30 | 89.34  |
| w/o  | 90.21    | 89.83 | 87.81 | 89.08 | 89.15  |

(b) Identification accuracy(%) with only one adversarial example.

|      | PGD-$\ell_2$ | C&W   | PGD-$\ell_\infty$ | APGD-CE | NES   | HSJA  | SignOPT |
|------|--------------|-------|-------------------|---------|-------|-------|---------|
| w/   | 68.98        | 80.89 | **85.56**         | 88.48   | **83.00** | **47.91** | **94.61**   |
| w/o  | **75.27**    | **83.74** | 76.24         | **89.15** | 73.79 | 45.36 | 91.45   |