# OpenReview forum: "An Implicit Watermark Framework for Adversary Identification"
_ICLR.cc/2024/Conference — Submitted to ICLR 2024_

### Official Review · Reviewer_iAzC · 2023-10-27

**Soundness:** 3 good
**Presentation:** 1 poor
**Contribution:** 2 fair
**Rating:** 3
**Confidence:** 3

**Summary:**

A recent study in adversarial ML is the forensic investigation of adversarial activity based on their generated adversarial samples. The goal is to identify an adversary based on the watermarked model copy they are attacking. By watermarking the model with specific architectural modules, the provider could identify the unique latent signature of the model through the adversary's query. However, previous studies have only considered the black-box attacker, whereby a provider is able to make architectural modifications to implement the watermarking. The authors propose a new framework where the adversary enjoys full knowledge of the model architecture, which is more general and challenging than previous work. This involves designing an implicit watermarking scheme that conceals model copy information into the model parameters without hurting performance. This is achieved in practice by fine-tuning model copies with blended images containing specific pixel-level fingerprints. These fingerprints are unique to each model copy and represent a set of coordinates where pixel information is overridden. The authors propose two metrics to identify the adversary's model copy with only one adversarial sample. For any given adversarial sample, the copy's fingerprint mask is applied to obtain the adversarially-perturbed fingerprint. From there the authors leverage the two metrics to capture low- and high-confidence attacks, and classify adversarial customers with around 70% accuracy on a single sample, and up to 100% accuracy with multiple samples.

**Strengths:**

* The manuscript studies a novel problem in large-scale distributed model deployment, which is identifying a bad actor among a pool of model copies. Compared to previous work, the authors study the harder white-box scenario.
* The paper is well organized and generally flows from section to section in a coherent manner.
* A framework for detecting adversarial customers could have positive impact to the model deployment community.
* The proposed watermarking framework has a minimal impact on the clean performance of the image classifier.
* The proposed identification metrics can correctly classify the adversary's model copy around 70% of the time with one sample, improving to up to 100% as more adversarial samples are included.
* The authors test against a strong white-box attack (APGD), older white-box attacks (PGD, CW), and black-box gradient estimation attacks (NES, HSJA).
* The authors re-implement the WILD and Neural Cleanse frameworks to check their proposed framework against watermark removal attacks. With only data augmentation, around 10% of cases are invalidated from WILD. By increasing the number of fingerprint classes to 4, Neural Cleanse is not able to detect the watermarks.

**Weaknesses:**

* The writing quality of the introduction and methodology sections is rather low, and some of the key concepts were difficult to interpret on a first pass. The writing quality improves as one delves deeper into the paper. There were some confusing word choices throughout the manuscript (e.g., water used interchangeably with watermark). The clarity of the early sections is low.
* Much of the notation could be cleaned up, particularly around Equation 2. The main issue is the number of repetitive parentheses. It also isn't clear why the authors choose $f_{att}$ when the model copies are denoted as $g$.
* The framework is designed with the assumption that multiple customers are performing the same classification task on the same training data, but this seems unrealistic to me. I would've liked to see the performance when different data is used to fine-tune customer models.
* The evaluation against adaptive attacks seems limited, as the authors only checked existing attacks and later adapt the watermarking hyperparameters, rather than consider the attack which targets the hyperparameters explicitly (see questions for the authors).
* Some design choices are unclear, such as needing the true label of the adversarial sample, or relying on the confidence delta of the output logits (see questions for the authors).
* The proposed framework is less effective against black-box adversaries, particularly attacks based on single-point gradient estimation (HSJA), which are the most realistic within the threat model.

**Questions:**

* The authors assume knowledge of the adversarial sample's true label, but it isn't clear how this information would be obtained. Are provider's expected to manually classify the adversarial samples with humans?
* It isn't clear where the proposed framework fits within a provider's deployment strategy. In other words, what are the events leading up to detecting the adversary's model copy? It is assumed that the adversarial sample was already found, but, related to the previous question, how does the provider know that the adversarial sample was crafted in the first place? My assumption is there would need to be some alert mechanism for detecting the adversarial sample, after which the framework is used. This isn't clarified in the main text.
* With respect to the framework design, it is assumed that a discrete coordinate set is used for obtaining the fingerprint, but can the adaptive adversary simply avoid perturbing this set of coordinates, and evade the provider?
* The adaptive attacks shed light on the case of watermark removal, but it seems more realistic that a white-box adversary would instead target the watermarking hyperparameters. Can the authors explain the scenario where the attacker knows the coordinate set, and actively tunes the sample against knowledge of the framework itself?
* The authors assume there are either low or high confidence attacks, and have proposed two metrics addressing either case. I suspect an adaptive adversary would design an attack so it always triggers one of the two metrics (i.e., trigger the least performant metric) to maximize their evasion likelihood. This partially explains the success of HSJA, since it does not focus on maximizing the softmax delta, instead only performing the single-point estimation at the decision boundary. In that sense, it seems the framework is too reliant on the presumed delta to exist. Can the authors elaborate on this possibility?
* Can the proposed framework work in the presence of multiple adversaries? Would it be possible for adversaries to cooperate amongst each other to obtain better evasion accuracy, by learning the watermark coordinates?

---

### Official Review · Reviewer_eSsX · 2023-10-29

**Soundness:** 2 fair
**Presentation:** 3 good
**Contribution:** 2 fair
**Rating:** 3
**Confidence:** 4

**Summary:**

This paper proposes a method for identifying the source model used to generate adversarial examples by placing a “watermark” on the model which is then carried over to the adversarial examples. The watermark is placed by essentially conducting a backdoor attack during the fine-tuning where each model is associated with a different random pattern.

**Strengths:**

### Originality

The problem setup is not completely new and almost identical to Cheng et al. [2023]. However, this work tries to address a more difficult variant where the adversary has full white-box access to the model being watermarked. I believe that this setting is a bit less realistic than Cheng et al. [2023]’s (only black-box access), but if one can solve this white-box setting, then the black-box setting is also solved.

### Clarity

The main ideas of the paper are conveyed clearly and effectively.

### Significance

The forensic problem is arguably less preferred than solving the adversarial robustness problem itself because it happens after the fact and the damage may have been done. However, it does provide some deterrence and an additional tool for dealing with the adversary. The paper also focuses on an interesting practical setting where there are many different copies of the models that are fine-tuned for the same task on the same dataset and share the same pre-trained model. This setting should apply to any company that provides a fine-tuning service.

**Weaknesses:**

### Weak experiment against backdoor defenses

To me, Section 4.3 is the most important in the paper; it addresses the largest concern of this empirical method. In particular, the assumption is that the adversary is (1) unaware of the backdoor pattern and is (2) unable to remove the backdoor. However, the experiment still falls a bit short for multiple reasons.

1. There is a rich literature on solving both of these problems as a defense against backdoor attacks: backdoor removal [1, 2, 3, 4] and backdoor detection [5, 6, 7]. It looks like the paper only covers one removal method and one detention method but does not explain why they were chosen instead of more recent and more powerful methods.
2. **Backdoor removal**. Generally, I would like to see more results to support the claim that the removal is ineffective when random erasing is used as data augmentation. This result on its own seems surprising and particularly important not only to this work but also to the broader backdoor attack/defense community. A few suggestions on how to strengthen this experiment:
    1. Explain why random erasing is effective.
    2. Do other data augmentation methods also lead to more robust backdoors?
    3. Does this conclusion hold against more sophisticated or more aggressive removal methods?
    4. Results on more datasets and models would make the argument more convincing as well.
3. **Backdoor detection**. What makes Neural Cleanse ineffective at detecting the backdoor? Why does using $|\mathbf y_i|=4$ instead of $2$ solve the problem? Have other detection methods been tried? Similarly to the backdoor removal result, I cannot find any table or figure that describes the quantitative results in this section at all.
4. If the state-of-the-art backdoor detection method still does not work, I may recommend a naive backdoor detection in this case: compute gradients of the loss w.r.t a test input (similar to how one generates adversarial examples), and then select the pixels with the top-k largest absolute gradients as the mask $m_i$. I would love to see the “accuracy” of multiple detection methods including this naive one.
5. Building on the previous point, it might be worth considering what the adversary could do to avoid being fingerprinted if he/she knows (1) both $m_i$ and $t_i$, (2) only $m_i$, and (3) only a noisy (or a subset) version of $m_i$. How effective would the adversary’s strategy be in each of these cases?
6. Lastly, the threat model should be stated more clearly, and there should be a discussion on the possibility of adaptive attacks, what their costs are, etc.

### Adaptive adversarial attacks

I believe that it is also important to consider adaptive attacks that do not involve removing or detecting the backdoor. Especially, the authors raise an interesting point on page 6 that “identification result on the black-box attack is not as good as white-box attack tested because of the noise gradient estimation used in the black-box attack.” This immediately suggests an adaptive attack that modifies any white-box attack by adding noise to the gradients. More generally, I wonder if methods for generating physically robust adversarial examples (i.e., expectation over transformation or EoT [8]) that apply random transformations to the input could be an effective adaptive attack. This is rather promising because attacks of this type are good at making the adversarial perturbation exploit more diverse and more robust features (i.e., lower frequency).

### Comparison to Cheng et al. [2023]

The authors note the reason for not comparing their scheme to Cheng et al. [2023] as “Cheng et al. [2023] is specifically designed for black-box attack identification which makes explicit modifications on the architectures and the white-box attacker could create strong adaptive attack to make the identification totally fail…” I do not believe that this reason does not invalidate the comparison under no adaptive attacker. If anything, this comparison could highlight the strength of the proposed scheme.

### False positive rate on attacks from non-watermarked models

I wonder what happens if the adversary uses a different (perhaps public) model not among the watermarked ones to generate adversarial examples. Since this paper formulates the identification task as a classification problem, one of the models will always be “blamed” for any given adversarial example. This can be extremely problematic so there must be an additional mechanism for determining if an adversarial example really comes from one of the fine-tuned models (e.g., detection threshold on the score).

### Other solutions to the same problem

As I understand, this paper focuses on fine-tuning services as an application where the adversary may use one of the many very similar copies of the model to mount a transfer to the other model copies. This is indeed an interesting and practically well-motivated setting, but I think there may be other defenses to solve the same problem.

For instance, if the transfer attack is the main concern here. The service provider may opt to deploy standard defenses against adversarial examples such as “adversarial training.” Other solutions include training methods that try to *diversify* each copy of the model [9, 10, 11]. These defenses against (transfer) attacks directly should be preferred to methods for identifying the culprit after the fact. I would like to see a detailed discussion that compares the pros and cons of different solutions.

### Clean accuracy vs identification accuracy trade-off

It should be possible to strengthen the watermark efficacy at the cost of the clean accuracy of the watermarked models. What are the knobs the service providers can use to tune this trade-off? Currently, the authors simply state the hyperparameter choices but do not clearly justify them. This experiment would be very interesting and should help strengthen the paper.

### Cost

I can see two main overheads of the proposed technique.

1. The score has to be computed over all the fine-tuned models so the cost scales with the number of models. In other words, the number of forward passes needed is $NM$ where $N$ is the number of models and $M$ is the number of adversarial examples being identified. $N$ can be anywhere from thousands to millions.
2. The ratio metric requires the ground-truth label. This means that there must be a human in the loop to provide this label.

While these costs should not completely prevent the deployment of this technique, I would still like to see these limitations mentioned and discussed in the paper.

### Ablation study

1. $T$ should be treated as a hyperparameter, and the result should be reported with the optimal choice of $T$. This can be tricky because a good value of $T$ also depends on the attack algorithm. There should exist a trade-off of identification accuracy among the attacks. There should be more experiments on how to choose a good value of $T$ in practice.
2. In the “effect of watermark design” paragraph, $T$ should be optimized for each strategy (discrete and square) instead of fixing it to the same value. There is no reason to believe that one value of $T$ will be good across different schemes.
3. I would like to see an ablation study on the choice of the identification metrics (KL and ratio). What is the accuracy if only one of the metrics is used at a time? What is the accuracy if other simpler schemes are used to average the two metrics?

- [1] https://arxiv.org/abs/1906.10842
- [2] https://arxiv.org/abs/2101.05930
- [3] https://proceedings.mlr.press/v139/shen21c.html
- [4] https://proceedings.mlr.press/v139/hayase21a.html
- [5] https://ieeexplore.ieee.org/document/9710346
- [6] https://arxiv.org/abs/2212.09067
- [7] https://arxiv.org/abs/2306.17441
- [8] https://arxiv.org/abs/1707.07397
- [9] https://arxiv.org/abs/2009.14720
- [10] https://arxiv.org/abs/2104.00671
- [11] https://proceedings.mlr.press/v162/rusu22a.html

**Questions:**

1. I wonder how the identification accuracy may change in the multi-sample case if the adversary uses different attacks for different samples. This should be fairly realistic for more sophisticated attackers.
2. Have the authors experimented with targeted attacks? I wonder how it affects the identification accuracy because it is likely that in the untargeted case, the attack will exploit the watermarked class $\tilde y_i$ and so the backdoored pattern. On the other hand, if the adversary uses a target label $y' \ne \tilde y_i$, the identification accuracy may be affected.

---

### Official Review · Reviewer_XRYs · 2023-10-30

**Soundness:** 1 poor
**Presentation:** 1 poor
**Contribution:** 2 fair
**Rating:** 3
**Confidence:** 2

**Summary:**

This paper attempted to identify the adversary from different model replicas in a white-box setting. Specifically, the authors fine-tuned each model copy using a slightly modified dataset, resulting in a unique watermark embedded in the parameters of each model. To identify the adversary, the authors compared the distribution of adversarial examples based on two similarity metrics. Experiments showed that the proposed method can recognize the adversary with a single adversary example.

**Strengths:**

$\bullet$ Compared with prior work, this work adopts a more challenging defense assumption.

$\bullet$ The proposed method recognized the adversary by adding watermarks to the model parameters, which is an interesting research area.

**Weaknesses:**

$\bullet$ The scenario and threat model of this work are not clear. In particular, what are the objectives and capabilities of both attackers and defenders? why are such attack and defense assumptions reasonable?
Indeed, from the defender's perspective, a white-box attack assumption is more challenging than the black-box setting employed in previous research, however, we should question whether it is necessary and reasonable to give attackers such extensive capabilities. The authors should elaborate more on scenarios where public access to the weights and architectures of the victim's or defender's models is necessary and justified. In other words, what is the rationale for requiring the architecture of the models to be publicly available? Moreover, in cases where the defender has identified the specific model being targeted for adversarial example generation, what should be the subsequent course of action? Without addressing these real-world issues and establishing a clear threat model, this work may lack practicality in real-world applications. The authors need to clarify the attack scenarios, describing their rationality and practical implications; clarify and formulate the goal of the attacker (e.g., to increase the attack success rate on a single model or to increase the transferability across multiple copies of a model), and the goal of the defender/recognizer.

$\bullet$ From a technical perspective, the approach proposed in this paper bears resemblance to the concept of a "Honeypot" employed for capturing adversarial attacks[Ref-1]. In [Ref-1], the authors deliberately introduce trapdoors, causing attackers' optimization algorithms to converge towards these trapdoors. The fingerprints developed in this paper represent another form of backdoors, which share similarities with the trapdoors described in [Ref-1]. Therefore, I find that the novelty within the technical aspect is somewhat restricted.

$\bullet$ How to ensure that watermarking has a negligible impact on the performance of each model replica? Table 1 shows that after fine-tuning, the classification accuracy of the model replicas decreased by 2%-5%.

$\bullet$ Why does the proposed method use a linear combination of two similarity metrics as the final similarity metric? Specifically, what is the physical significance of the Ratio metric proposed in this paper, and what is the physical significance of the linear combination of the KL distance and the Ratio metric?

$\bullet$ Experiments lack comparisons with baseline methods. For example, Table 1-4 only presents the performance of the proposed method without comparison with baseline methods.

$\bullet$ Some of the experimental settings are confusing. For example, in Section 4.1, why was it applied using 0.9% of the total number of images, and in Section 4.2, why was the threshold T set to 7?

[Ref-1] Shan, Shawn, et al. "Gotta catch'em all: Using honeypots to catch adversarial attacks on neural networks." Proceedings of the 2020 ACM SIGSAC Conference on Computer and Communications Security. 2020.

**Questions:**

A more detailed description of why and how fingerprint adversarial examples are generated is needed.

---

### Official Review · Reviewer_BRji · 2023-11-01

**Soundness:** 3 good
**Presentation:** 2 fair
**Contribution:** 2 fair
**Rating:** 5
**Confidence:** 4

**Summary:**

In this paper, the author put forward a framework to trace the compromised model under the white-box adversarial attack. The authors first constructed different fingerprint samples for each copy of the baseline model and implicitly watermarked these copies by fine-tuning them. Given one or more adversarial copies, the authors then computed a scalar score to evaluate every copy, which is a linear combination of KL matric and ratio matric. The copy with the lowest score is identified as the compromised copy.

**Strengths:**

(1) The idea of identifying the network vulnerable to adversarial attack under the white-box setting is interesting.

(2) The experiments are overall comprehensive. The authors demonstrated the limited degradation of classification accuracy of the watermarked model and the identification accuracy in both white-box and black-box attack settings.

**Weaknesses:**

(1) The assumption of the threat model is strong. The proposed framework is only useful when there is one and only one adversary among all model copies. This is a strong limitation. It is possible that adversarial examples come from multiple adversaries, in which case the framework can still identify only one of them.

(2) More evidence is needed to support the claimed assumption. (i) For example, the assumption for the KL metric, “when the model predicts x_adv with high confidence on the fingerprint class y_j”, then“the generated adversarial perturbation would be very similar with the sampled pixels t_i” is not well supported. (ii)Also, the assumption for the ratio metric, “we observed that there would be a significant change on the model prediction distribution after applying t_i for the model that x_adv is based”, is not well supported. After applying t_i to the clean model, wouldn’t there also be a change in the distribution? How do g_att and clean model g_i differ from each other?

(3) Claims for the robustness of the proposed watermark are not well supported. For the fine-tuning-based methods, it’s claimed that simple data augmentation will make the proposed watermark more robust. However, when users fine-tune their copy with their own data and various data augmentation methods, how are original watermarks ensured not to be removed? A study of the impact of fine-tuning a clean model is needed. Even if there is only one adversary, the normal users’ copies of the model can also be fine-tuned, but the impact is not studied.

(4) For the reverse-engineering-based methods, it’s claimed that increasing the number of fingerprint classes will make these methods unable to detect watermarks. However, a study on the sensitivity of the outlier's threshold for the picked method, Neural Cleanse[a], is needed. Also, the claim, “the number of fingerprint class |y_i| has limited effect on the identification accuracy and it can even further improve identification accuracy” needs more experiments to be supported. Only the results with 4 fingerprint classes are reported in Table 3 and Table 4.

(5) Some notations are not well defined. Such as m_{a,b,c} in section 3.2.

[a] Bolun Wang, Yuanshun Yao, Shawn Shan, Huiying Li, Bimal Viswanath, Haitao Zheng, and Ben Y Zhao. Neural cleanse: Identifying and mitigating backdoor attacks in neural networks. In 2019 IEEE Symposium on Security and Privacy (SP), pp. 707–723. IEEE, 2019.

**Questions:**

(1) Is it possible that the adversarial example is misclassified into another class (y_k) other than the fingerprint class (eg. y_adv) of g_att with high confidence? In this case, is it true that a clean model g_i other than g_att, which happens to have the fingerprint class y_k, will be misidentified as the adversary by your proposed KL metric?

(2) The authors mentioned, “we simply increase the number of fingerprint class |y_i| = 4” in section 4.3. Does that mean, for every model g_i, the investigator has multiple triggers, each corresponding with a different fingerprint class?

(3) For the experiments based on CIFAR-10, when the number of copies is 50, this means that at least 5 copies have the same fingerprint class, suppose this class is y_k. These 5 copies have 5 different sampled pixels t_i. Suppose the model predicts x_adv with high confidence on the class y_k and the g_att is one of these 5 copies. How do you ensure that the generated adversarial perturbation is very similar to one of these t_i and is very different from the other four t_i?

(4) What if the adversary fine-tuned their copy with another set of fingerprint samples, such that the compromised model can have multiple different fingerprints? Will this copy still have a low KL metric or ratio metric when evaluated with the original fingerprint?

(5) Does the framework always identify one and only one adversary, even when there are multiple adversaries or no adversaries?

(6) “Our identification result on the black-box attack is not as good as the white-box attack tested because of the noise gradient estimation used in the black-box attack.” Why is the identification accuracy for black-box attacks the best in Figure 2 (b) when the number of adversarial examples is greater than one, better than all white-box attacks?

(7) The choice of T. Since every copy is dispatched to different users and the copies can be modified arbitrarily, every copy can have a different confidence score, how can the investigator decide T? Is one T chosen for all copies, or do different copies have different T values? If T is not chosen properly, how can one tell if a misidentification happens? For example, a normal user with a low-accuracy model is identified as the adversary.

---

### Meta-Review · Area_Chair_cYZF · 2023-12-10

**Metareview:**

While the reviewers appreciated the paper’s motivation, experimental setup, and clarity of the writing, their main concerns were with (a) missing comparisons (especially with respect to Cheng et al., 2023), (b) lack of clarity on the threat model, (c) overheads for applying the method, and (d) missing details explaining design choices. There was no author response. For these reasons I vote to reject. The reviewers have given extremely detailed feedback and I recommend the authors follow / respond to their comments closely before submitting to a future ML venue. If the authors are able to fix these things it will make a much stronger submission.

**Justification For Why Not Higher Score:**

All reviewers argued reject and there was no author response, should have been withdrawn.

**Justification For Why Not Lower Score:**

N/A

---

### Decision · Program_Chairs · 2024-01-16

Reject